# Comparing Devices for Concurrent Measurement of AC Current and DC Injection during Electric Vehicle Charging †

**Olga Mironenko** [1],* and **Willett Kempton** [1,2]

[1]  Department of Electrical and Computer Engineering, University of Delaware, Newark, DE 19716, USA; willett@udel.edu

[2]  College of Earth, Ocean and Environment, University of Delaware, Newark, DE 19716, USA

*  Correspondence: olgamiro@udel.edu

†  The manuscript is based upon the first author's Ph.D. thesis, chapter 4.

**Abstract:** Widespread adoption of electric vehicles (EVs) requires additional safety countermeasures to prevent DC injection from EVs into the AC grid via Electric Vehicle Supply Equipment (EVSE). Moreover, for energy purchase, and even more so for vehicle-to-grid (V2G) services, the EVSE must conduct high precision bidirectional power and energy measurements. This paper introduces operating principles, structure, performance, and cost comparison of three current sensing technologies—current transformer, shunt and fluxgate—for metering and protection within an EVSE, concluding with recommendations among those sensors for the most beneficial applications concerning EV charging.

**Keywords:** EV; V2G; EVSE; DC injection; fluxgate; shunt; current transformer; EV charging; revenue metering; current measurement

## 1. Introduction

Currently, transformerless topology inverters experience high interest due to their smaller size, higher efficiency, and lower cost compared to traditional inverters which have an isolation transformer at their output. However, any transformerless faulty power converter can inject some portion of DC current into the AC grid. This effect is called "DC injection". Acceptable level of DC injection is specified in the "Limitation of DC injection" section of the IEEE 1547-2018 standard [1]. The level of DC injection above 0.5% of the full rated output current can cause various negative effects to other equipment, such as saturation and overheating transformers and AC motors, higher losses, and acceleration of cable corrosion in the grounding wires [2–4]. It is especially important for high power converters such as employed in electric vehicles (EVs). Therefore, this article recommends that the level of DC injection from EVs should be monitored via electric vehicle supply equipment (EVSE) as a gateway between the grid and EV's power converters. A separate check in the EVSE would be an economically efficient way to ensure grid safety. Due to regulatory officials being especially concerned about bidirectional charging, verification of DC injection for Vehicle-to-Grid technology (V2G), which enables the bidirectional power flow between the electric vehicle (EV) batteries and the grid, may be considered especially important [4,5].

The EVSE governs the connection of grid power to the vehicle power converter, and thus is a major part of any EV system. Some EVSEs are capable of measuring both the instantaneous power flow and the accumulated kWh of energy consumed. The measurement may be required for EVSE working in conjunction with grid measurements such as controlled charging. Furthermore, V2G capable EVSE can

allow conducting power back to the grid, and thus, the metering must be bidirectional. The authors hypothesize that it would be cost-efficient to use the same device to measure both the AC charging power and the DC injection. However, the two orders of magnitude differences of measured AC vs. DC currents makes using the same instrument challenging. Fortunately, it is the much larger AC quantities that require greater accuracy. Despite the rapid growth of EV charging, extensive literature search found no sources evaluating current sensors for this dual purpose. For this literature review see [4].

The accuracy criteria for power and energy and, consequently, AC current measurements for revenue meters (so-called "revenue grade" accuracy) are defined in NIST HB-44, ANSI C12.20 and IEC 62053-11 standards (0.5, 1.0 and 2.0 accuracy classes) [6–8]. A meter will satisfy the majority of regional transmission operators (RTOs) in North America and Europe, if the meter is able to meet all three accuracy classes. However, U.S. markets such as PJM are satisfied with meter's performance within 1.0 or 2.0 accuracy classes [9].

The system requirements for the current sensing system inside the V2G capable EVSE are summarized in Table 1.

**Table 1.** Current metering system requirements for single-phase and for three-phase EVSEs.

| Parameters | Requirements | |
|:---:|:---:|:---:|
| | **Single-Phase (19.2 kW)** | **Three-Phase (100 kW)** |
| DC injection limit | 0.5% of $I_{rated}$ | |
| DC injection detection accuracy at the limit value | 20% | |
| AC current metering accuracy classes | 0.5, 1.0, 2.0 | |
| $I_{rated}$ | 80 A | 120 A |

The system must obey the standards for AC revenue metering accuracy and for DC injection limitation (see Section 1). Please note that "Limitation of DC injection" section of IEEE 1547-2018 standard does not specify an accuracy of DC injection detection. Rather, the designer must establish sufficient margin so that the greatest likely error will still be unlikely to be more than the required threshold value. Based on measurements of accuracy of DC injection reported in [4], the authors estimate that a 20% margin will generate false errors only for injection within 20% but will very rarely fail to detect over-standard DC injection. In addition, the system must satisfy the AC current range of $I_{rated}$ as well as must be compatible with electrical systems worldwide (as much as practical).

This paper compares operating principles, performance, and cost (Prices are collected from Digi-Key Electronics, Mouser Electronics, Vango Technology, KG Technologies and Magnetic Sensor Systems (estimate for 1000+ quantity). Parts price below $0.5 is not included.) of three current sensing systems: current transformer (CT), shunt, and fluxgate for single-phase and three-phase EVSE in application of high precision AC revenue metering and DC injection detection. In addition, the paper recommends the most optimal solution for concurrent revenue metering and DC injection detection according to the system requirements listed in a Table 1 as well as possible EV charging applications for other two systems.

## 2. Current Transformer Current Sensing System

### 2.1. Operating Principle

The most common AC current sensing system for revenue-grade metering uses a current transformer (CT). Since it is based on Faraday's Law of induction, CT can only detect the AC current, it is blind to DC injection.

A CT consists of a ferrite core and primary and secondary windings coupled together so that the induced current in the secondary coil is proportional to the primary current. The former can be calculated from the induced current using the known windings turn ratio. In power measurement

devices, the primary "winding" is typically the main current-carrying wire passing through once or looped a few times around the CT coil containing the secondary winding (see Figure 1).

CTs are popular for revenue-grade metering systems due to their high accuracy, low complexity, and isolation. In addition, CTs are directly compatible with analog-to-digital converters. However, high accuracy CTs can be costly, as they employ high relative permeability core materials [4,10–12].

### 2.2. System Structure and Capabilities

The CT current sensing system (Figure 1) was designed and developed at the University of Delaware by the V2G research group. The UD-developed system is currently in use for high precision AC current, voltage and power measurements in EVSE commercial units.

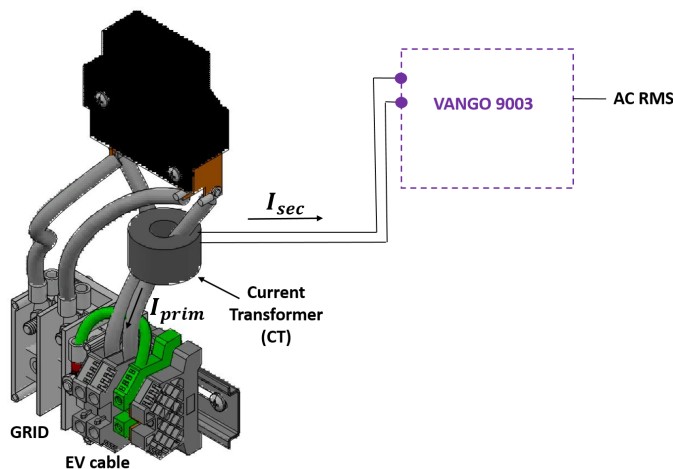

**Figure 1.** Current transformer current sensing system for a single-phase EVSE.

The system consists of the BCT-013-200 CT, ±0.1% precision resistor network and the Vango 9003 metrology chip, used in many submeters. The CT is placed around the main current conductor, and the other elements are integrated into a circuit board.

The current to be measured $I_{prim}$ is picked up by CT. Secondary current $I_{sec}$, proportional to $I_{prim}$, is induced and processed through the metrology chip, where the current RMS value is calculated. The system is specified to measure AC currents up to 200 A.

The main disadvantage of the system is its incapability to detect DC currents (see Section 2.1).

### 2.3. Accuracy

The manufacturer guarantees the BCT-013-200 CT is ±0.1% accuracy. Therefore, the calibration of each EVSE unit is not required, which simplifies the EVSE manufacturing process. The overall CT sensing system has been measured and certified by TESCO Engineering to 1.0 accuracy class.

### 2.4. Cost

The detailed cost of the single and three-phase systems is shown in Table 2.

As we mentioned in Section 2.2, the CT current sensing system is only capable of detecting an AC current. Therefore, the separate DC current sensor must be installed to extend the measurement range to DC currents. The authors selected a closed-loop Hall-Effect current sensor LEM LA 150-P with operating range up to 150 A and 0.5% accuracy at the primary nominal current of 150 A. In addition, the price of dual power supply LT1945EMS, required for the sensor, was included in the cost estimate.

**Table 2.** Current transformer current sensing system: detailed cost for single and three-phase EVSE. Hall-Effect LEM sensor is added only for DC injection detection price estimate.

|  | Single-Phase System | | Three-Phase System | |
|---|---|---|---|---|
| Part No | Qty | Price | Qty | Price |
| BCT-013-200 CT | 1 | $14 | 3 | $42 |
| Vango 9003 | 1 | $3.15 | 1 | $3.15 |
| LEM LA-150P | 1 | $19.04 | 3 | $57.12 |
| LT1945EMS | 1 | $2.60 | 1 | $2.60 |
| **Total cost** | | $38.79 | | $104.87 |

The closed-loop Hall Effect sensor belongs to magnetic sensors family and is capable of measuring AC and DC signals. The sensor's operating principle is based on Hall Effect when the Lorenz force affects the charged particle moving in the presence of a magnetic field.

The closed-loop Hall Effect sensor concentrates the magnetic field created by the current to be measured in a magnetic core surrounding the main conductor. This magnetic field is measured through the Hall voltage. The output Hall voltage is converted to the secondary current which is proportional to the current under interest [4,13,14].

Please note that the Hall-Effect sensor was not tested for the ability to detect 400 mA DC injection; it is presented here for the price comparison, not necessarily as a technology ready for this purpose. In addition, it is important to mention, that the authors do not include a Hall-Effect current sensor as a stand alone solution for the application of interest due to its lack of standards-required accuracy at low currents.

## 3. Shunt Current Sensing System

### 3.1. Operating Principle

A shunt is based on Ohm's Law of resistance and thus is capable of measuring both AC and DC currents. The measured voltage drop across the shunt is proportional to the current flowing through it. A shunt is a simple, low-cost current sensing solution with an acceptable level of accuracy. Typical measuring accuracy of a shunt is $\pm 0.1\%$, $\pm 0.25\%$, or $\pm 0.5\%$. Drawbacks of shunt current sensors are (1) the absence of galvanic isolation, (2) the heat release due to power dissipation on a resistor, and (3) the weak output signal due to a voltage drop in several mV ranges, requiring signal amplification [4]. For more details regarding shunt operating principle and related concerns, such as energy losses at high currents, see [4].

### 3.2. System Structure and Capabilities

The shunt current sensing system prototype was also designed and developed by the University of Delaware V2G research group [4]. The prototype system structure is shown in Figure 2.

The system consists of a single shunt current sensor and two readout circuits (AC and DC). The shunt is installed into a latching safety relay lead inside the EVSE. The current to be measured generates a voltage drop across the shunt. The voltage drop is picked up by readout circuits and processed by Vango 9003 metrology chip.

The circuit on the top is responsible for high precision AC current measurements, whereas the circuit underneath is dedicated to DC injection detection. In order to determine whether the DC injection value is above the safe limit or not, Vango 9003 splits the bottom analog circuit's AC + DC mixed output signal by its components and calculates the DC injection. Low-pass filter in DC injection circuit requires approximately 100 ms to settle. Vango 9003 AC and DC measurements are updated once per 20 ms and stabilized in 80 ms.

The system is verified to be linear for AC current from 6 A to 80 A. In addition, it can simultaneously detect DC injection of 400 mA or more in an AC signals up to 80 A. However,

the system has been tested only for a 19.2 kW single-phase EVSE, it has a potential to be extended to a 100 kW, 120 A three-phase EVSE. Detailed test platform configuration and simulated EV load profile information can be found in [4].

Please note that in the next design revision, all the components will be integrated into the main printed circuit board (PCB), but due to time and fabrication cost constrains, the prototype presented in this paper is designed on a separate PCB board connected to the main one.

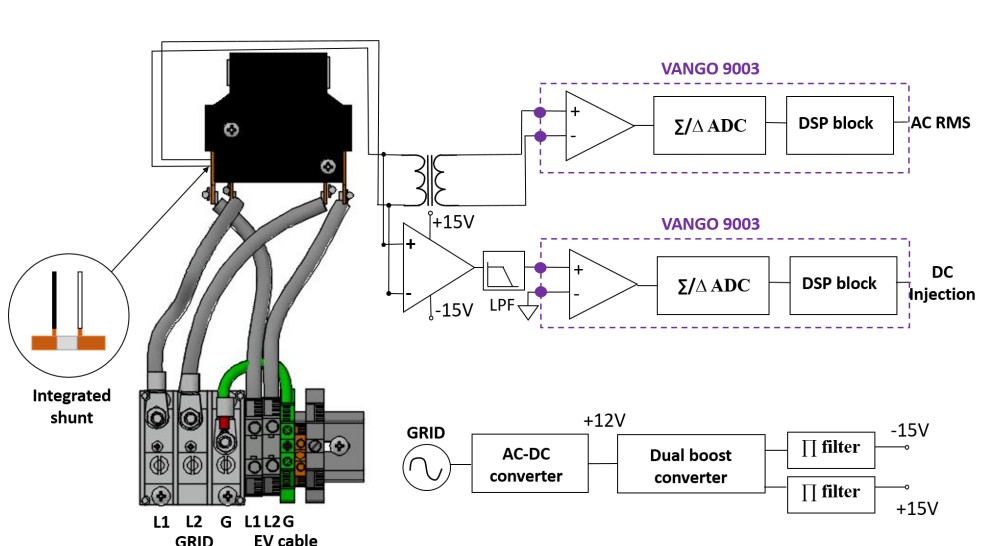

**Figure 2.** Shunt current system's prototype for single-phase EVSE.

### 3.3. Accuracy

For AC current revenue metering the system meets 1.0 accuracy class requirements defined in NIST HB-44 when charging (power flows from the grid to an EV) and when discharging (power flows from an EV to the grid).

The accuracy of DC injection detection at the limit value of 400 mA is 8.25% when DC injection is positive, and the accuracy is 16.5% when DC injection is negative. Moreover, we believe, that more precise calibration and the prototype integration into the main PCB will result in further accuracy improvement. Detailed accuracy results of measured AC current and DC injection can be found in [4].

Although the shunt current sensing system fulfills the requirements listed in Table 1, calibration of each EVSE unit is essential due to ±5% shunt tolerance. In addition, further work is required to integrate the prototype into commercial EVSE production.

### 3.4. Cost

The cost of the system is shown in Table 3. (Because some components are not simply three units for the three-phase EVSE (DC/DC converter) or result from a single manufacturing change (three-phase relay), the three-phase price is not necessarily three times the single-phase price.)

Table 3 lists prices for the major parts of the system. In addition to shunt sensor and Vango 9003 metrology chip, authors included prices for the signal transformer TTC-5036 used in AC readout circuit, operational amplifier AD8479ARZ-RL used in DC readout circuit and DC/DC converter LT1945EMS used in dual boost converter (Figure 2).

**Table 3.** Shunt current sensing system: detailed cost for single and three-phase EVSE.

| | Single-Phase System | | Three-Phase System | |
|---|---|---|---|---|
| **Part No** | **Qty** | **Price** | **Qty** | **Price** |
| Shunt | 1 | $3 | 3 | $5 |
| Vango 9003 | 1 | $3.15 | 1 | $3.15 |
| TTC-5036 | 1 | $2.52 | 3 | $7.56 |
| AD8479ARZ-RL | 1 | $4.25 | 3 | $12.75 |
| LT1945EMS | 1 | $2.60 | 1 | $2.60 |
| **Total cost** | | $15.52 | | $31.06 |

## 4. Fluxgate Current Sensing System

### 4.1. Operating Principle

Fluxgate current sensors are magnetic field sensors that can detect low-frequency AC and DC currents, in contrast to CT. Fluxgate uses a non-linear relation between the ferromagnetic material permeability and the ambient magnetic field.

The basic fluxgate sensor consists of a ferromagnetic rod-shaped core, an excitation coil, and a pick-up coil. The high-frequency excitation signal is sent through the excitation winding to drive the core to saturation in both directions. At the moment of core saturation, a significant reduction in its magnetic permeability occurs, causing the ambient magnetic flux to collapse. The magnetic flux change induces a measurable voltage in the pick-up coil, which is proportional to the ambient magnetic field. The Fluxgate sensor's name derives from the magnetic flux "gating" effect at the saturation point. Although fluxgate is one of the most accurate magnetic sensors (up to $\pm0.0002\%$), their application is limited due to its complexity and high price of commercially available products [4,15,16]. More details about fluxgate operating principle and common fluxgate topologies can be found in [4].

### 4.2. System Structure and Capabilities

The Fluxgate current sensing system prototype was developed by Magnetic Sensor Systems LLC [17]. The system consists of a fluxgate sensor head, placed around a copper conductor, and a controller board. The controller board has excitation and readout circuits for a sensor head.

The current to be measured flows through the conductor, generating a magnetic flux picked up by a sensor head, and processed by a controller board (Figure 3).

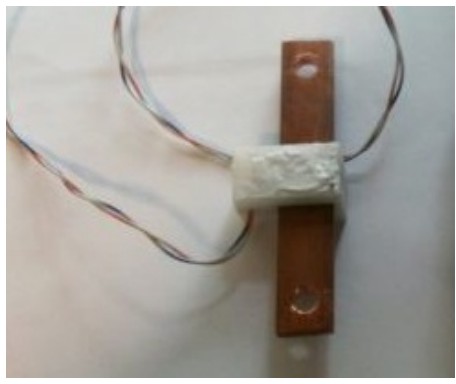
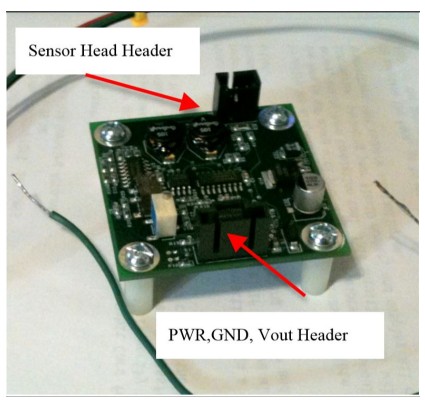

(**a**)   (**b**)

**Figure 3.** Fluxgate current sensing system's prototype: (**a**) The sensor head. (**b**) Controller board (images used by permission of Magnetic Sensor Systems, LLC).

Due to its nature, the fluxgate sensor current sensing system is capable of conducting both AC measurements and DC injection detection. The system is able to measure AC and DC currents in the range of ±50 A. In addition, the fluxgate sensor head is highly consistent across several production units. Therefore, no calibration per unit is required in serial production.

Please note that the system has been tested at the minimum current of 1 A. Therefore, further testing is required to conclude on the sensor's ability to detect DC injection ≥400 mA. In addition, as it has been mentioned above, the Fluxgate current sensing system is on a prototype stage, and further design work is needed to incorporate the current sensing system into any production device, such as an EVSE.

*4.3. Accuracy*

Accuracy data for DC currents is provided by Magnetic Sensor Systems LLC [17]. For positive currents, the system demonstrates the accuracy about 0.1% of full scale (FS) for currents up to 30 A. However, accuracy is lower at higher currents, where the accuracy is 1.2% of FS near 50 A. For negative currents, the system is accurate to approximately 0.3% of FS.

*4.4. Cost*

The detailed cost of the system is shown in Table 4.

**Table 4.** Fluxgate current sensing system: detailed cost for single and three-phase EVSE.

|  | Single-Phase System | | Three-Phase System | |
| --- | --- | --- | --- | --- |
| **Part No** | **Qty** | **Price** | **Qty** | **Price** |
| Fluxgate sensor head | 1 | $0.57 | 3 | $1.71 |
| Controller board | 1 | $14.85 | 3 | $44.55 |
| Vango 9003 | 1 | $3.15 | 1 | $3.15 |
| **Total cost** | | $18.57 | | $49.41 |

The cost estimate for each part of the fluxgate sensing system is provided by Jeff Viola, Magnetic Sensor Systems LLC. (Email from 10/17/2019, Jeff Viola, MSS llc, jeff@magsensorsystems.com) Please note that Vango 9003 metering chip is included only to estimate the approximate cost of the sensor's output signal processing.

## 5. Systems Comparison and Recommended Applications

The comparison of aforementioned systems is presented in Table 5.

**Table 5.** Comparison of three current sensing systems for single-phase and three-phase EVSEs.

| Characteristics | CT + Hall Effect | Fluxgate | Shunt |
| --- | --- | --- | --- |
| System complexity | low | high | moderate |
| Operating current range | 150 A | 50 A | 80 A (verified) 120 A (in progress) |
| Operating voltage range | Isolated | Isolated | 600 V |
| AC revenue metering accuracy class (0.5,1.0,2.0): | 1.0 | N/A | 1.0 |
| DC accuracy, positive (@ 400 mA) | N/A | N/A | 8.25% |
| Dc accuracy, negative (@ 400 mA) | N/A | N/A | 16.5% |
| Calibration/unit | not required | not required | required |
| Total cost, single phase EVSE | $38.79 | $18.57 | $15.52 |
| Total cost, three phase EVSE | $104.87 | $49.41 | $31.06 |

A CT current sensing system can be successfully used for high precision revenue metering inside the EVSE. However, the DC current is invisible for CT. Therefore, the CT system cannot be employed

for DC injection detection without an additional DC current sensor. Consequently, overall cost of the system will increase approximately twice (see Section 2.4).

In contrast, the fluxgate current sensing prototype system is capable of detecting DC current and has an affordable price in comparison with commercially available fluxgate current sensors. However, the system has not been tested at the currents below 1 A.Therefore, further testing is required to evaluate the sensor's ability to detect DC injection with required accuracy. In addition, the system has limited operating current range. Therefore, we do not recommend the system at its current state for a single-phase or a three-phase V2G EVSEs, specified in a Table 1.

The shunt current sensing system can detect DC injection within the EVSE system requirements (Table 1) and conduct bidirectional AC current measurements with acceptable accuracy for 19.2 kW single-phase EVSE.The authors verified accuracy up to 80 A, the shunt system is believed able to extend the measurement range up to 120 A three-phase EVSE. Moreover, the shunt system is cost-effective compared to the other three systems considered in the paper, especially for three-phase application. Therefore, the shunt sensing system holds promise for V2G capable as well as regular EVSE applications. Although the system is currently on a prototype stage, we believe that it can be successfully incorporated into EVSE units for commercial production.

## 6. Conclusions

In this work, we compared operating principles, structure, performance and the cost of CT, fluxgate, and shunt current monitoring systems for concurrent AC revenue metering and DC injection measurements for EV charging and discharging. We described advantages and weaknesses of each system, and provided a recommendation for their most beneficial applications with EV charging. The shunt current sensing system was recommended as the best suited for simultaneous detection of DC injection and accurate AC revenue metering [4]. As the authors hypothesized, the shunt system is the most cost-effective.

**Author Contributions:** Conceptualization, O.M. and W.K.; methodology, O.M.; formal analysis, O.M.; investigation, O.M. and W.K.; writing—original draft preparation, O.M.; writing—review and editing, W.K.; project administration, W.K.; funding acquisition, W.K. All authors have read and agreed to the published version of the manuscript.

**Funding:** This research was funded by a grant from Nuvve Corp. to the University of Delaware. Research Agreement, 1 September 2019.

**Acknowledgments:** The authors would like to thank Fouad Kiamilev from University of Delaware for providing laboratory space and for valuable advice.

**Conflicts of Interest:** The authors declare no conflict of interest.

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
