# Peer review of "Comparing Devices for Concurrent Measurement of AC Current and DC Injection during Electric Vehicle Charging†"

_wevj, doi:10.3390/wevj11030057_

Round 1

Reviewer 1 Report

The paper introduced operating principles, structure, performance and cost comparison of three current sensing technologies for or metering and protection within an EVSE. A detailed comparison among three current sensing technologies---current transformer, shunt current and fluxgate current monitoring systems---for concurrent AC revenue metering and DC injection measurements during EV charging and discharging is provided by the authors. I think the paper is well organized, and each current monitoring scheme is clearly demonstrated, however, further improvements should be made in the revised manuscript and my suggestions are as follows:

  1. The authors estimate a safe tolerance to be 10%-20% of the limit value. I think more information, such as literature and test result, etc., needs to be provided to support this estimation.
  2. It is highly recommended to add the test platform configuration and simulated EV load profile information for the monitoring experiment. Also, a comparison between measured and reference values needs to be added and the values should be plotted under different to support the claimed accuracy results.
  3. For any monitoring or protection system, the response time is also an important metric to evaluate the performance. To give the readers a comprehensive comparison, the authors are suggested to provide related information from the developed sensing system or the literature survey.
  4. There are always trade-offs between cost and performance. Based on the comparison of three current sensing systems for single-phase and three-phase EVSEs, discussions on how to properly select these types of current sensing systems regarding the application requirements are highly recommended to be included in this paper.

Author Response

  1. The authors estimate a safe tolerance to be 10%-20% of the limit value. I think more information, such as literature and test result, etc., needs to be provided to support this estimation.

The authors clarified the selection of a 20% limit value and provided a reference to the test results to justify the given number. The authors also replaced the range of 10%-20% to 20% limit to avoid confusion and misunderstanding.

L 48-53: "Rather, the designer must establish sufficient margin so that the greatest likely error will still be unlikely to be more than the required threshold value. Based on measurements of accuracy of DC injection reported in [6], the authors estimate that a 20% margin will generate false errors only for injection within 20% but will very rarely fail to detect over-standard DC injection."

      2. It is highly recommended to add the test platform configuration and simulated EV load profile information for the monitoring experiment. Also, a comparison between measured and reference values needs to be added and the values should be plotted under different to support the claimed accuracy results.

The authors provided a reference to the test platform configuration, simulated EV load profile information, and comparison between measured and reference values.

L 136-137: "Detailed test platform configuration and simulated EV load profile information can be found in [6]."

L 148: "Detailed accuracy results of measured AC current and DC injection can be found in [6].

Instead of a plot, the authors demonstrated the accuracy results in table 3 (see ref [6]). The table better represents the results since the reference/measured current values vary from 8A to 35A whereas measurement errors are within 1% of a reference current. Therefore, such a small deviation won't be clearly visible on a plot, and therefore, demonstrated in a table.  

      3. For any monitoring or protection system, the response time is also an important metric to evaluate the performance. To give the readers a comprehensive comparison, the authors are suggested to provide related information from the developed sensing system or the literature survey.

The authors provided the required information on timing from the developed sensing system.

L 130-132: "Low-pass filter in DC injection circuit requires approximately 100 ms to settle. Vango 9003 AC and DC measurements are updated once per 20 ms and stabilized in 80 ms. "

     4. There are always trade-offs between cost and performance. Based on the comparison of three current sensing systems for single-phase and three-phase EVSEs, discussions on how to properly select these types of current sensing systems regarding the application requirements are highly recommended to be included in this paper.

Discussion is already presented under section 5.

Reviewer 2 Report

The paper compares three current sensing systems including; current transformer (CT), shunt, and fluxgate for single-phase and three-phase EVSEs for DC injection detection. The reviewer has the following comments;

  • In the Introduction section, please provide further details about the necessity of sensing DC injection for the case of EV’s integration in power systems. For example, what are the origins and impacts of DC injection considering different charging levels and charging equipment.

  • Further explanations, e.g. the mathematics for the working principle, specifically regarding shunt and fluxgate current sensing systems can be added.

Author Response

  • In the Introduction section, please provide further details about the necessity of sensing DC injection for the case of EV’s integration in power systems. For example, what are the origins and impacts of DC injection considering different charging levels and charging equipment.

The requested information is provided.

L 12-19: "Currently, the transformerless topology inverters experience high interest due to their smaller size, higher efficiency, and lower cost compared to traditional inverters having an isolation transformer at their output. However, any transformerless faulty power converter can inject some portion of DC current into the AC grid.
This effect is called ”DC injection”. Acceptable level of DC injection is specified in the "Limitation of DC injection" section of the IEEE 1547-2018 standard [2]. The level of DC injection above 0.5% of the full rated output current can cause various negative effects to other equipment, such as saturation and overheating transformers and AC motors, higher losses, and acceleration of cable corrosion in the grounding wires"

  • Further explanations, e.g. the mathematics for the working principle, specifically regarding shunt and fluxgate current sensing systems can be added.

The authors provided a reference to more detailed explanation of the operating principles, common topologies, and concerns of shunt and fluxgate technologies.

L 118-119: "For more details regarding shunt operating principle and related concerns, such as energy losses at high currents, see [6]."

L 170-171: "More details about fluxgate operating principle and common fluxgate topologies can be found in [5]."

Reviewer 3 Report

In their work, the authors compare three different sensing systems to measure AC and DC injection during the charging process of electric vehicles. Each sensing system is analyzed for their suitability for bidirectional charging based on different previously defined parameters. The authors conclude that among all three systems, the shunt sensing system is best suited for the described purpose.

The authors should posit how their work contributes to the literature and how it is novel. To this end, a summary of the current state of literature should be added as well. 

The authors are analyzing and comparing the three systems for different requirements. The reviewer proposes that a short section is added for each of the systems that include criteria such as durability and longevity since these can affect long-term costs.

The references are not formatted consistently; should be changed accordingly.

Detailed Comments:

  • L. 14: The authors should shortly list examples of the adverse effects
  • L. 24: The authors should remove the apostrophe for the plural
  • L. 27: A comma is missing after “thus.”
  • L. 29: The authors should state why it is expected to be cost-effective to use the same device for both measurements
  • L. 39: The “a” should be removed 
  • L. 46-47: Table 1 - The authors mention the requirement of low costs as an important criterion for the devices. However, a clear definition of what is considered low costs is missing.
  • L. 78: A reference should be added that defines CT systems as inaccurate
  • L. 83: Subsection 2.3.1 should be changed to 2.4 to keep a uniform numbering system
  • L. 96: References should be listed in chronological order
  • L. 110: References should be listed in chronological order
  • Table 3: The costs for the shunt with a quantity of 3 should be three times the costs for a single shunt. The given price does not seem to be correct.
  • L. 158: The authors highlight the high price of fluxgate current sensing systems. In fact, the price for this system is lower compared to the comparable CT for both single-phase and three-phase systems. 
  • L. 185: For a better understanding, the reviewer proposes that the authors focus on CT currents sensing systems that include a DC sensor since only this option is comparable to the other sensing systems. The costs of this comparable version are then higher than the costs for the other systems and cannot be described as inexpensive.
  • L. 194-200: The authors should mention the cost-effectiveness of the shunt current sensing systems compared to the other systems

Author Response

The authors should posit how their work contributes to the literature and how it is novel. To this end, a summary of the current state of literature should be added as well. 

L 35-37: "Despite the rapid growth of EV charging, extensive literature search found no sources evaluating current sensors for this dual purpose. For this literature review see [5]"

The authors are analyzing and comparing the three systems for different requirements. The reviewer proposes that a short section is added for each of the systems that include criteria such as durability and longevity since these can affect long-term costs.

Since two systems (shunt and fluxgate) are currently in a prototype stage, the data about durability and longevity is not available.

The references are not formatted consistently; should be changed accordingly.

The authors accommodated this requirement.

Detailed Comments:

  • L. 14: The authors should shortly list examples of the adverse effects

L 18-19: "as saturation and overheating transformers and AC motors, higher losses, and acceleration of cable corrosion in the grounding wires"

  • L. 24: The authors should remove the apostrophe for the plural

Removed

  • L. 27: A comma is missing after “thus.”

Comma is added

  • L. 29: The authors should state why it is expected to be cost-effective to use the same device for both measurements

The authors replaced "it would be expected" to "hypothesize". The hypothesis is confirmed by the data.

L 31-32: "The authors hypothesize that it would be cost-efficient..."

L 222-223: "As the authors hypothesized, the shunt system is the most cost-effective."

  • L. 39: The “a” should be removed 

Removed

  • L. 46-47: Table 1 - The authors mention the requirement of low costs as an important criterion for the devices. However, a clear definition of what is considered low costs is missing.

The "low cost" entry is removed from the manuscript (L 55)

  • L. 78: A reference should be added that defines CT systems as inaccurate

The inaccuracy statement is removed from the manuscript since it doesn't affect the comparison results. (L 86)

  • L. 83: Subsection 2.3.1 should be changed to 2.4 to keep a uniform numbering system

Changed

  • L. 96: References should be listed in chronological order
  • L. 110: References should be listed in chronological order

Accommodated

  • Table 3: The costs for the shunt with a quantity of 3 should be three times the costs for a single shunt. The given price does not seem to be correct.

The price origins are clarified in a footnote to table 3.

L 153: "Because some components are not simply three units for the three-phase EVSE (DC/DC converter) or result from a single manufacturing change (three-phase relay), the three-phase price is not necessarily three times the single-phase price."

  • L. 158: The authors highlight the high price of fluxgate current sensing systems. In fact, the price for this system is lower compared to the comparable CT for both single-phase and three-phase systems. 

The authors clarified it in the text.

L 170: "high price of commercially available products"

L 202-203: "In contrast, the fluxgate current sensing prototype system is capable of detecting DC current and has an affordable price in comparison with commercially available fluxgate current sensors."

  • L. 185: For a better understanding, the reviewer proposes that the authors focus on CT currents sensing systems that include a DC sensor since only this option is comparable to the other sensing systems. The costs of this comparable version are then higher than the costs for the other systems and cannot be described as inexpensive.

Accommodated. (L 198)

  • L. 194-200: The authors should mention the cost-effectiveness of the shunt current sensing systems compared to the other systems

Accommodated.

L 211-213: " Moreover, the shunt system is cost-effective comparing to the other three systems considered in the paper, especially for three-phase application."

Round 2

Reviewer 1 Report

The authors considered the review’s suggestion and in the revised manuscript, more detailed clarifications and additional information are provided, which are greatly appreciated.

At my standing point, this paper is expected to be accepted. The authors are encouraged to proofread the paper regarding wording, grammar, and formats before submission.